# Disease Acceptance and Control from the Subjective Health Experience Model as Health Perception Predictors in Immune-Mediated Inflammatory Diseases

**DOI:** 10.3390/biomedicines13030538

**Published:** 2025-02-20

**Authors:** Tessa S. Folkertsma, Sjaak Bloem, Robert M. Vodegel, Reinhard Bos, Greetje J. Tack

**Affiliations:** 1Department of Dermatology, Medisch Centrum Leeuwarden, MCL—Medical Center Leeuwarden, Henri Dunantweg 2, 8934 AD Leeuwarden, The Netherlands; r.vodegel@mcl.nl; 2Department of Rheumatology, Medisch Centrum Leeuwarden, MCL—Medical Center Leeuwarden, Henri Dunantweg 2, 8934 AD Leeuwarden, The Netherlands; r.bos@mcl.nl; 3Department of Gastroenterology, Medisch Centrum Leeuwarden, MCL—Medical Center Leeuwarden, Henri Dunantweg 2, 8934 AD Leeuwarden, The Netherlands; greetje.tack@mcl.nl; 4Center for Marketing & Supply Chain Management, Nyenrode Business University, 3621 BG Breukelen, The Netherlands; s.bloem@nyenrode.nl

**Keywords:** immune-mediated inflammatory diseases (IMIDs), immunology, subjective health experience, health perception, personalised supportive healthcare, patient-reported outcomes, structural equation modelling, psoriasis (PsO), rheumatoid arthritis (RA), inflammatory bowel disease (IBD)

## Abstract

**Background/Objectives:** The multifactorial nature of immune-mediated inflammatory diseases (IMIDs) requires integrating pathophysiological understanding with subjective patient experiences. Patient-reported outcome measures (PROMs) are a useful tool for incorporating this in routine clinical practice. However, current PROMs do not fully encompass the complete subjective health experiences (SHE) of patients and are thus insufficient for guiding truly personalised care. The SHE model provides insights into SHE through the determinants of disease acceptance and perceived control. While validated across various demographics, its predictive power in IMIDs cohorts remains unexplored. This study aims to assess whether acceptance and perceived control in the SHE model predict health experiences in patients with IMID and how these immunological conditions compare. **Methods:** Questionnaires regarding health perception, acceptance, control, and subjective health experiences were distributed to 450 Dutch citizens. Descriptive statistics, reliability checks, and partial least squares structural equation modelling were used to examine relationships between variables. **Results:** Health perception strongly predicts SHE through acceptance and control. Across all conditions, the pathway moves from health perception to control, then to acceptance, and finally to SHE. However, the roles of acceptance and control differ by condition. In burdensome diseases like inflammatory bowel disease and rheumatoid arthritis, acceptance plays a greater role, while control has a stronger influence in conditions like psoriasis. **Conclusions:** This study supports the predictive validity of the SHE model for IMIDs, showing that disease acceptance and control affect health experiences differently across conditions. These insights improve the understanding of psychological factors in health experiences and call for tailored interventions for patients with IMID.

## 1. Introduction

The impact of immune-mediated inflammatory diseases (IMIDs) extends beyond their physical manifestations, profoundly affecting quality of life (QoL) [1,2]. While the physical complications of these diseases are well documented, ranging from cardiovascular issues in rheumatoid arthritis (RA) [3] and psoriasis (PsO) [4] to fistulas [5] and malnutrition [6,7] in inflammatory bowel disease (IBD), the psychological impact is equally significant. Individuals with IMID often struggle with depression, anxiety, stigmatisation, and social withdrawal, underscoring the need for comprehensive care strategies [8,9,10]. The multifactorial nature of these conditions requires the integration of pathophysiological understanding with subjective patient experiences. A patient’s overall subjective perception of how illness and treatment impact their physical, psychological, and social well-being is referred to as health-related QoL (HRQoL) [11]. Despite advances in treatment, managing IMIDs remains challenging due to variations in disease mechanisms and individual responses to therapy. Emerging research highlights that while IMIDs share common inflammatory pathways, differences in cytokine function and immune regulation contribute to variability in treatment effectiveness [12]. Many individuals remain unresponsive or intolerant to current therapies, underscoring the need for novel approaches that target multiple inflammatory pathways simultaneously.

The use of patient-reported outcome measures (PROMs) could be a useful tool for incorporating HRQoL in daily healthcare [13]. However, their application into routine clinical practice is limited in its suitability as a tool for tailoring patient-specific care. Transforming PROMs results into effective, personalised interventions remains challenging. Broekharst et al. highlight this in their study of a cohort of IBD patients [14], emphasising the need to integrate a general HRQoL measure alongside traditional disease activity assessments. According to Broekharst et al., while current PROMs (like the EuroQol Five-dimensions Five-level; EQ-5D-5L) reflect health perception, they do not fully encompass the complete HRQoL. This distinction is critical, as health perception primarily addresses how patients perceive specific aspects of their illness, such as pain, limitations, or emotional distress, but it falls short of capturing the broader, subjective experience of health itself. As a result, current PROMs are insufficient to guide truly personalised care.

Bloem and Stalpers developed a model that does provide insights into the complete Subjective Health Experience (SHE) of patients [15]. They characterise SHE as an individual’s physical and mental functioning, aligned with their desired lifestyle, yet constrained by the realities of their existence. SHE goes beyond health perception, capturing not only how patients perceive their illness but also how they subjectively experience their overall health. Therefore, SHE provides a more accurate representation of HRQoL. For example, two individuals with the same disease burden may experience vastly different levels of subjective health, with one perceiving their condition as far worse than the other. Stalpers found that in chronically ill patients, including patients with asthma, rheumatoid arthritis, various types of cancer, and psoriasis, the most important psychological determinants of subjective health were disease acceptance and perceived control [16]. Increased acceptance, indicative of a patient’s ability to integrate their health condition into everyday life, combined with a sense of control, reflecting their capacity to improve their health, positively influences SHE. In the SHE model, the evaluation of disease acceptance and perceived control involves three specific questions each, making the model simple and highly applicable in routine clinical practice.

The SHE model has previously demonstrated its versatility and has been validated across various patient demographics and disease domains [17,18]. In addition, the SHE model has recently been adapted to address the specific needs of patients with IMID [19]. Despite the previously demonstrated value of the SHE model in healthcare, the predictive power of the model has yet to be examined in immunological patient cohorts. Therefore, this study aims to determine whether the disease acceptance and control parameters of the SHE model can predict experiences of health and illness in patients with IMID (IBD, PsO, and RA) and whether or not these conditions show similarities. The findings from this study could further legitimise and support the integration of the SHE model for the easy evaluation of health experiences in routine clinical practice for patients with IMID.

## 2. Materials and Methods

This study is part of an extensive research project aiming to offer and monitor more personalised supportive care related to IMIDs and treatments based on PROMs. The study protocol was reviewed by the accredited Independent Ethics Committee and approved by the Board of Directors of the Medical Centre Leeuwarden (RTPO, nWMO 20200091, January 2021). This study utilised a cross-sectional research design featuring online questionnaires. An overview of the research process is presented in Figure 1. The questionnaires were distributed to a panel of Dutch citizens by the market research firm IPSOS between February and December 2023. To tailor distribution, IPSOS used responses from a general pre-screening questionnaire to identify potential participants with predefined diagnoses. Questionnaires were then sent until a predetermined quota was reached. Panel members received comprehensive information about the study and were asked to provide formal consent prior to participation. Only respondents who consented to the use of their data for research purposes and who reported an official diagnosis (by a physician) of IBD (Crohn’s disease or ulcerative colitis), PsO, or RA were included in the study. Although spondyloarthropathy was initially included as a diagnosis in the study, the number of respondents was too small for meaningful analysis, and therefore, it was excluded from the final sample. The final sample consisted of 450 Dutch participants. IPSOS supplied the full dataset, including only those who completed the survey in its entirety.

Questionnaires included various items on sample characteristics as well as PROMs. Demographics, including age, gender, residential region, area population density, education level, annual income, and time since diagnosis, were used to describe the study population. These items were measured on nominal scales containing binary response categories, as well as ordinal scales containing ascending response categories. The EQ-5D-5L questionnaire was used to measure health perception [20]. The dimensions (i.e., mobility, self-care, usual activities, pain and discomfort, and anxiety and depression) were measured with a five-point scale. The SHE questionnaire was used to determine acceptance, control, and SHE [16]. Acceptance and control were measured using three items equipped with a seven-point Likert scale ranging from 1 = fully disagree to 7 = fully agree. SHE was measured as the average perception of health for the previous month by marking a ladder with 11 levels, in which level 0 indicates the worst day of the previous month and level 10 indicates the best day of the previous month.

The characteristics of the study population were analysed using descriptive statistics, with percentages reported for categorical variables and means for continuous variables. Cut-off values were based on statistical data from the general Dutch population [21,22,23]. Partial least squares structural equation modelling (PLS-SEM) was employed to determine the relationships between the instruments (subsamples: 5000). Criteria for the reliability and validity of the instruments, as well as model interpretation, followed the guidelines outlined by Hair et al. [24]. The construct reliability of the instruments was assessed using Cronbach’s alpha (α), rho_A, and rho_C coefficients and was deemed reliable if their values exceeded 0.70. Construct validity was assessed by establishing convergent and discriminant validity. Convergent validity was examined using the average variance extracted (AVE) coefficient and considered sufficient if it surpassed 0.50. Discriminant validity was analysed using the heterotrait–monotrait (HTMT) ratio, which was sufficient if below 0.90, and the Fornell–Larcker (FL) criterion, which was satisfied if the AVE of a construct was greater than its correlation with any other construct. Effect sizes were determined using standardised beta coefficients (β), which were classified as small if below 0.30, average if between 0.30 and 0.50, and large if above 0.50. Significance levels were assessed using *p*-values, which were considered significant if below 0.05. Explained variance was analysed using R-squared (R^2^) coefficients, which were considered small if below 0.30, average if between 0.30 and 0.50, and large if above 0.50. IBM SPSS Statistics Version 28 was used for describing the final sample, and SmartPLS Version 4.1 was deployed for analysing the instruments and models.

## 3. Results

### 3.1. Participant Characteristics

For this study, 450 panel members were included, comprising 100 individuals with IBD (50 with Crohn’s disease and 50 with ulcerative colitis), 200 with PsO, and 150 with RA. Although SpA was initially included as a diagnosis, the number of respondents was too small to allow for a meaningful analysis. As a result, SpA was excluded from the final analysis. The cohort resembled the general Dutch population in terms of the residential region and population density (Table 1). However, the population was relatively older, slightly more female-dominated, and had a lower education level and lower income [25]. These cohort characteristics are, however, representative of the characteristics of individuals with IMID [26,27]. The differences in sample sizes between diagnoses were due to limitations in the availability of participants within the IPSOS panel. Despite this, the patient demographics were relatively comparable across diagnoses.

### 3.2. Instrument Characteristics

To verify the accuracy and credibility of the models, the reliability and validity of the instruments were determined. The instruments of health perception (α = 0.78–0.82; rho_A = 0.79–0.84; rho_C = 0.85–0.87), acceptance (α = 0.89–0.90; rho_A = 0.90–0.92; rho_C = 0.93–0.94), and control (α = 0.90–0.93; rho_A = 0.90–0.93; rho_C = 0.93–0.96) demonstrated reliability, with coefficients exceeding 0.70. The instruments of health perception, acceptance, and control had sufficient construct validity, as all their items loaded on a single component. The instruments of health perception (AVE = 0.53–0.59), acceptance (AVE = 0.81–0.83), and control (AVE = 0.82–0.88) demonstrated sufficient convergent validity, with coefficients exceeding 0.50. The instruments of health perception (HTMT = 0.50–0.74), acceptance (HTMT = 0.57–0.83), control (HTMT = 0.50–0.83), and SHE (HTMT = 0.50–0.79) demonstrated sufficient discriminant validity, as coefficients remained below 0.90, and the FL criterion was satisfied.

### 3.3. Model Characteristics

The model for predicting SHE with health perception via acceptance and control was first examined for the total study population, followed by the IBD, PsO, and RA subsets.

#### 3.3.1. Total Study Population Model

The total IMID sample model (Figure 2; Model 1) showed statistically significant relationships between all variables. Health perception was a strong predictor (β = 0.59, *p* < 0.01) of control, which was a strong predictor (β = 0.52, *p* < 0.01) of acceptance, which was a strong predictor (β = 0.52, *p* < 0.01) of SHE. The effect of health perception on acceptance (β = 0.28, *p* < 0.01) and control on SHE (β = 0.19, *p* < 0.01) was considered small. The model explained 35%, 51%, and 44% of variance in control, acceptance, and SHE, respectively, which is considered moderate to large.

#### 3.3.2. IBD Model

The IBD sample model (Figure 2; Model 2) demonstrated statistically significant relationships between all variables except for the relationship between control and SHE. Again, health perception was a strong predictor of control (β = 0.63, *p* < 0.01), which was a strong predictor of acceptance (β = 0.52, *p* < 0.01), which was a strong predictor of SHE (β = 0.68, *p* < 0.01). The effect of health perception on acceptance (β = 0.23, *p* < 0.01) and control on SHE (β = 0.10, *p* = 0.41) was considered small. The model explained 39%, 48%, and 56% of variance in control, acceptance, and SHE, respectively, which is considered moderate to large.

#### 3.3.3. PsO Model

The PsO sample model (Figure 2; model 3) demonstrated statistically significant relationships between all variables. As with previous findings, health perception was a strong predictor of control (β = 0.55, *p* < 0.01), and acceptance was a strong predictor of SHE (β = 0.52, *p* < 0.01). However, in this sample, the effect of health perception on acceptance (β = 0.37, *p* < 0.01) and control on acceptance (β = 0.40, *p* < 0.01) were considered moderate. The effect of control on SHE (β = 0.25, *p* < 0.01) was considered small. The model explained 30%, 46%, and 49% of variance in control, acceptance, and SHE, respectively, which is considered moderate.

#### 3.3.4. RA Model

The RA sample model (Figure 2; model 4) showed statistically significant relationships between all variables, except for the relationship between control and SHE. As with the PsO sample, health perception was a strong predictor of control (β = 0.62, *p* < 0.01), which in turn strongly predicted acceptance (β = 0.62, *p* < 0.01). The effect of acceptance on SHE was moderate (β = 0.41, *p* < 0.01). The effects of health perception on acceptance (β = 0.23, *p* < 0.01) and control on SHE (β = 0.17, *p* = 0.41) were considered small. The model explained 38%, 60%, and 31% of variance in control, acceptance, and SHE, respectively, which is considered moderate to large.

## 4. Discussion

The aim of this study was to determine whether the disease acceptance and control parameters of the SHE model can predict experiences of health and illness in patients with IMIDs (restricted here to IBD, PsO, and RA) and whether or not these conditions show similarities. This study demonstrates that disease acceptance and perceived control are key psychological mechanisms mediating the relationship between health perception and subjective health experience (SHE) in patients with IMID, with notable variations in the roles of these factors across different conditions.

The total study population model indicates that the impact of health perception, measured via the EQ-5D-5L questionnaire, on SHE through disease acceptance and control is strong within this population, confirming the significant relationships between all variables. This means that the relationship between health perception (specifically the effect of illness and intervention) and SHE is mediated by the psychological variables of disease acceptance and perceived control over disease. This result aligns with the findings of Stalpers [15], confirming their earlier conclusions regarding the importance of these factors in mediating SHE.

Acceptance and control emerge as crucial mechanisms for optimising SHE, a pattern that is consistent with studies by Bloem et al. and Broekharst et al. [16,17], further underscoring the reliability of these constructs in health outcomes. Notably, we also observe a significant indirect effect, where perceived control influences SHE via acceptance, highlighting the importance of perceived control in indirectly shaping health experiences through its impact on acceptance. This indirect effect will be explored in more detail in subsequent sections.

When analysing the PsO model, a similar trend to the total study population model emerges. Both acceptance and control play pivotal roles in predicting SHE. However, in PsO, the influence of acceptance and control is slightly more balanced. This suggests that while the route from perception to SHE remains significant, the roles of control and acceptance are more proportionate, making their combined influence more synergistic.

In contrast to PsO, the models for IBD and RA reveal a distinct pathway in which acceptance predominantly mediates SHE, with no significant direct relationship between control and SHE. This confirms earlier work by van Erp [28], which established a relationship between control and acceptance in IBD and is further supported by Teugels et al. [29]. Our findings extend this understanding by showing that the same pattern applies to RA, indicating that in both IBD and RA, acceptance plays a central mediating role. However, while Teugels focused only on correlations, our study goes beyond, testing a more comprehensive model that also accounts for indirect effects. In this context, control is indeed related to SHE but indirectly through its influence on acceptance. This provides a more nuanced understanding of the relationship between control and SHE in IBD, contrasting with Teugels’ conclusions.

One possible explanation for the more prominent role of disease acceptance in IBD and RA compared to PsO is that the disease burden is generally higher in gastroenterological (including IBD) and rheumatological conditions (including RA) [17]. These variations in acceptance and control levels could explain the proportional effect sizes (β) observed in the models. As Broekharst et al. noted, the greater disease burden may require IBD and RA patients to expend more energy or time to achieve higher levels of acceptance, which in turn influences SHE [17]. Further research is needed to confirm this assumption.

Additionally, PsO patients may have access to more treatment options or perceive greater support, which could directly improve their SHE without requiring as much psychological adaptation via acceptance. Further research is currently underway to explore this hypothesis in greater detail.

Looking at the models and the causal pathway with the strongest effects, a common pattern is observed across all models: health perception influences perceived control, which in turn affects acceptance, ultimately leading to SHE. This suggests that interventions aimed at improving patients’ perceived control could directly contribute to better acceptance and, ultimately, better SHE. Supporting this, our recent qualitative study examined the type of information and support that patients with IBD, RA, and PsO need for their treatment [19]. The study found that support, tailored for improving both acceptance and control, can play a crucial role in optimising SHE, reinforcing the importance of these psychological variables in patient care. 

Generally, improving acceptance tends to take more time than improving control. Furthermore, support for enhancing control is typically more practical and instrumental, such as providing information about the diagnosis and defining the treatment plan. In contrast, improving acceptance requires more patient-centred guidance. Given this, focusing on control in the initial stages of treatment may be more time-efficient and immediately beneficial. Further research is currently exploring the relationship between the time since diagnosis and levels of acceptance and control, potentially identifying whether the pathways in the SHE model differ across varying stages of diagnosis.

A key strength of this study lies in the high reliability and validity of the measurement instruments used, which enhances the accuracy and credibility of the findings. Furthermore, the use of partial least squares structural equation modelling for this study instead of multiple correlations has a number of advantages. Structural equation modelling accounts for the measurement error of latent variables and avoids inadmissible solutions and factor indeterminacy [30,31].

Another strength is the ability to show meaningful relationships within the SHE model, even with smaller sample sizes, as demonstrated in this study. Although the effect of control on SHE for IBD and RA was not significant, the significance of models with larger sample sizes suggests that this issue could be addressed by increasing the sample size.

However, there are some limitations to consider. The final sample may be biased due to a relatively older age group, a slight overrepresentation of females, and lower education levels among respondents, which could potentially distort or misrepresent the findings. Additionally, this study’s focus on a Dutch population may limit the interpretability and generalisation of the results to other contexts.

The findings of this study have important implications for clinical practice. They suggest that the SHE model is applicable within different types of IMIDs, making it a useful general tool for managing the treatment of multiple immune-mediated disorders. The simplicity of the model makes it well-suited for clinical settings. Furthermore, the results indicate that improving health perception can be achieved by specifically targeting the levels of acceptance and control perceived by patients, offering a strategic approach to enhancing patient outcomes. In the initial stages of treatment, the focus should be on improving control rather than acceptance for disorders with a high disease burden.

At least four main directions for future research may be pursued. The first direction focuses on further segmenting this model based on other important demographic variables, such as annual income and character traits. Stalpers found in an exploratory study that these factors could influence the model, but it remains to be explored whether this holds true within the IMID population [15].

The second direction involves validating the model using disease-specific PROMs to assess health perception. Since these PROMs are more tailored to the individual patient, they may provide a more accurate representation of their health perception, potentially improving the model’s predictive accuracy.

The third direction highlights the need for validation of this model in various countries to better understand the impact of geographical location and cultural differences on its applicability.

The fourth direction centres on integrating this model into clinical practice. By using the model to personalise healthcare, the goal would be to improve health perception through targeted strategies that focus on disease acceptance and control.

## 5. Conclusions

In conclusion, while health perception consistently predicts SHE across all disorders, the roles of acceptance and control vary between IMIDs, with acceptance playing a larger role in more burdensome diseases like IBD and RA and control having a more direct influence in conditions like PsO. These insights not only confirm existing findings but also contribute to a deeper understanding of how psychological factors mediate health experiences in patients with IMID.

## Figures and Tables

**Figure 1 biomedicines-13-00538-f001:**
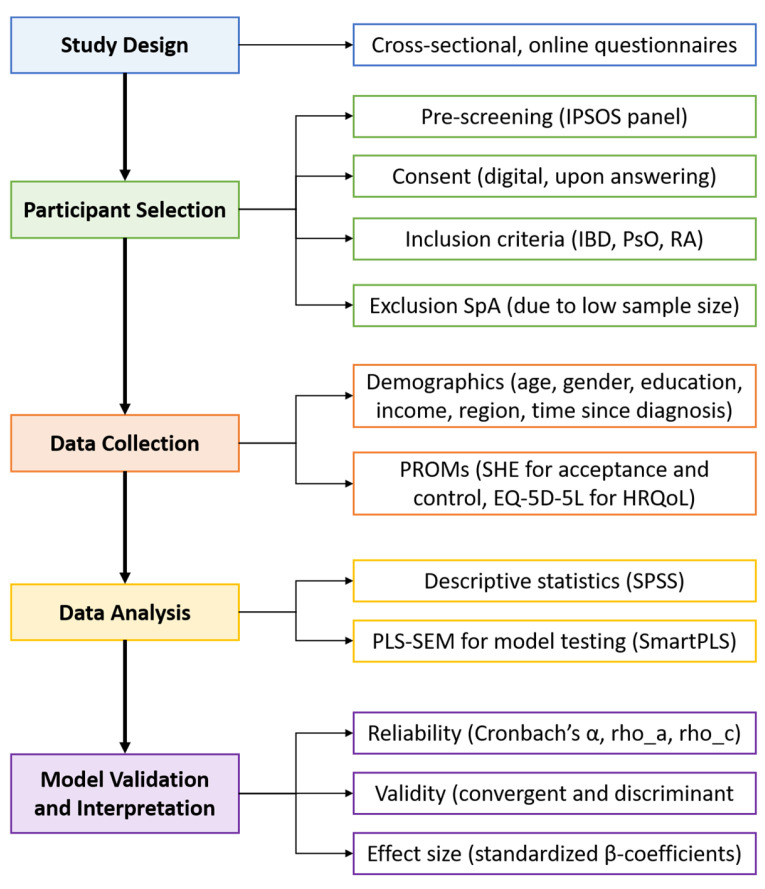
Analytical process chart.

**Figure 2 biomedicines-13-00538-f002:**
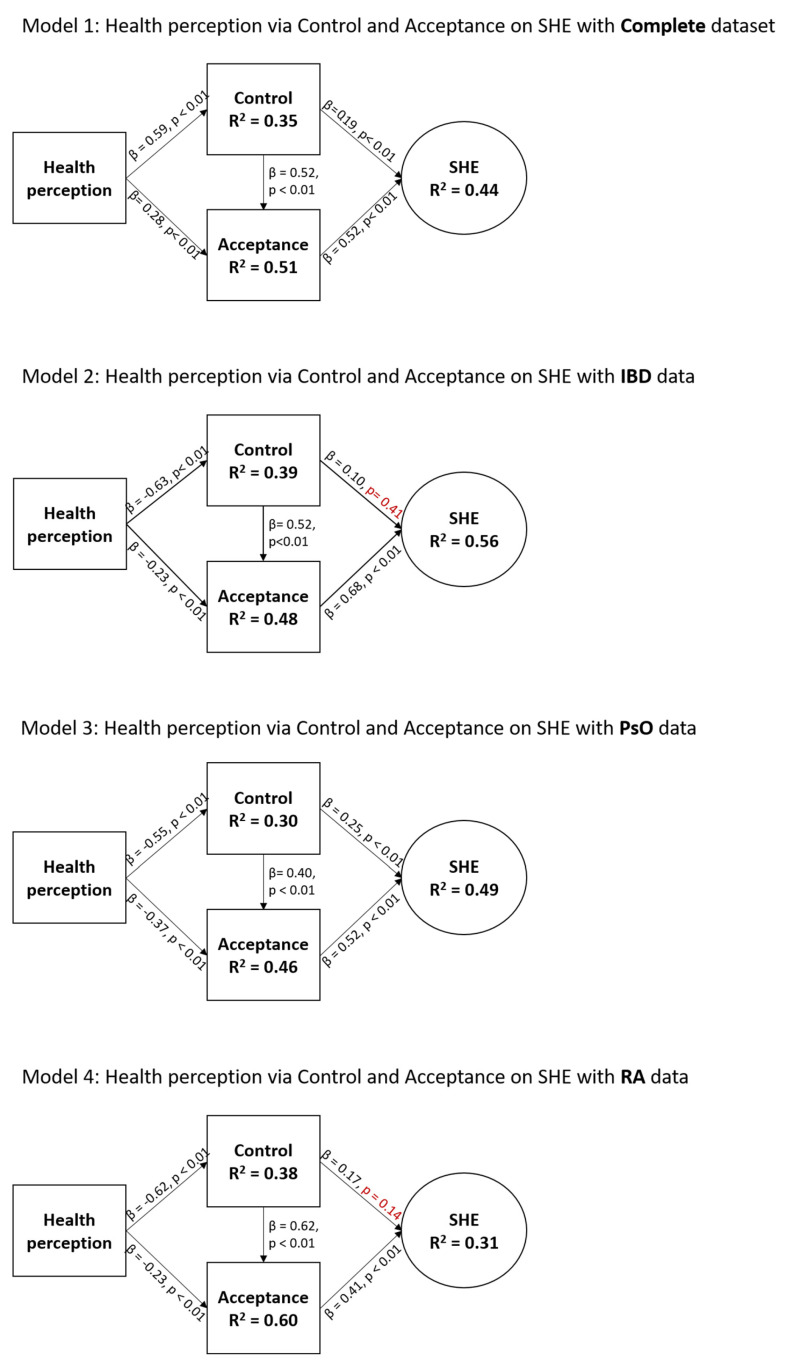
Characteristics of the SHE model for the total study population and for disease-specific data. Health perception was measured via the EQ-5D-5L questionnaire, and acceptance, control, and SHE were measured via the SHE questionnaire. Non-significant *p*-values (*p* > 0.05) are indicated in red.

**Table 1 biomedicines-13-00538-t001:** Sample characteristics of the total study population and per diagnosis.

	Total Study Population	*n* = 450		
Age	61.0 years (22–87 years)			
Gender	58.9% Female	41.1% Male		
Residential region	35.3% North and East NL	40.7% West NL	24.0% South NL	
Area population density	12.0% ≤ 200 inhab/km^2^	40.7% 200–999 inhab/km^2^	47.3% ≥ 1000 inhab/km^2^	
Education level *	28.4% Low	38.9% Middle	32.4% High	
Annual income	49.5% < EUR 36,500	17.0% EUR 36,500–EUR 43,500	33.5% > EUR 43,500	
Time since diagnosis	36.7% 0–10 years	21.4% 11–20 years	18.1% 21–30 years	23.8% > 30 years
	**IBD**	*n* = 100		
Age	57.1 years (30–81 years)			
Gender	53.0% Female	47.0% Male		
Residential region	39.0% North and East NL	30% West NL	31.0% South NL	
Area population density	9.0% ≤ 200 inhab/km^2^	42.0% 200–999 inhab/km^2^	49.0% ≥ 1000 inhab/km^2^	
Education level *	23.0% Low	40.0% Middle	37.0% High	
Annual income	37.4% < EUR 36,500	20.4% EUR 36,500–EUR 43,500	42.2% > EUR 43,500	
Time since diagnosis	25.0% 0–10 years	29.0% 11–20 years	23.0% 21–30 years	23.0% > 30 years
	**PsO**	*n* = 200		
Age	61.1 years (22–83 years)			
Gender	53.0% Female	47.0% Male		
Residential region	35.0% North and East NL	43.5% West NL	21.5% South NL	
Area population density	17.5% ≤ 200 inhab/km^2^	39.5% 200–999 inhab/km^2^	43.0% ≥ 1000 inhab/km^2^	
Education level *	25.5% Low	37.5% Middle	37.0% High	
Annual income	46.2% < EUR 36,500	18.5% EUR 36,500–EUR 43,500	35.3% > EUR 43,500	
Time since diagnosis	32.4% 0–10 years	21.4% 11–20 years	12.1% 21–30 years	34.1% > 30 years
	**RA**	*n* = 150		
Age	63.5 years (32–87 years)			
Gender	70.7% Female	29.3% Male		
Residential region	33.3% North & East NL	44.0% West NL	22.7% South NL	
Area population density	6.7% ≤ 200 inhab/km^2^	41.3% 200–999 inhab/km^2^	52% ≥ 1000 inhab/km^2^	
Education level *	36.2% Low	40.3% Middle	23.5% High	
Annual income	61.9% < EUR 36,500	12.7% EUR 36,500–EUR 43,500	25.4% > EUR 43,500	
Time since diagnosis	50.7% 0–10 years	15.9% 11–20 years	22.5% 21–30 years	10.9% > 30 years

* Low education level: primary education level, VMBO, the first three years of HAVO/VWO, and entrance training, including the former assistant training (MBO1). Middle education level: upper levels of HAVO/VWO, basic vocational training (MBO2), vocational training (MBO3), or middle management and specialist training (MBO4). High education level: education at higher vocational or university level.

## Data Availability

The raw data used during this study are not publicly available but are available from the corresponding author upon reasonable request.

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
