# Peer review of "Disease Acceptance and Control from the Subjective Health Experience Model as Health Perception Predictors in Immune-Mediated Inflammatory Diseases"

_biomedicines, 2025, doi:10.3390/biomedicines13030538_

Round 1
Reviewer 1 Report
Comments and Suggestions for Authors
In this study, the authors assessed whether disease acceptance and control in the SHE-22 model predict health experiences in patients with immune mediated inflammatory 15 diseases (IMID). This analyze could further support the the SHE-model for IMID, showing that disease acceptance and control affect the health experiences differently across conditions. This is an interesting study with important scientific significance and clinical values. Several minor comments should be addressed before the accept.
An analytical process chart should be added as an image to highlight the logical structure of the research.
There are some grammatical and spelling errors in the paper. Please carefully review the manuscript and make the necessary corrections.
Can the authors provide more Color Figures to make the data more substantial and beautiful.
The emerging studies about immune mediated inflammatory diseases also should introduced, such as doi: 10.1186/s40001-024-01836-1.
Comments on the Quality of English LanguageThere are some grammatical and spelling errors in the paper. Please carefully review the manuscript and make the necessary corrections.
Author Response
Comment 1: An analytical process chart should be added as an image to highlight the logical structure of the research.
Answer 1: We sincerely appreciate your valuable suggestion. To enhance the clarity of the research’s logical structure, we have added a new figure (Figure 1, page 3, line 118) that visually represents the framework of our study.
Comment 2: There are some grammatical and spelling errors in the paper. Please carefully review the manuscript and make the necessary corrections.
Answer 2: Thank you for bringing this to our attention. We have carefully reviewed the manuscript to ensure grammatical accuracy and correct any spelling errors. Additionally, we have uploaded a revised version with all modifications highlighted as a supplementary file for transparency.
Comment 3: Can the authors provide more Color Figures to make the data more substantial and beautiful.
Answer 3: We appreciate your feedback. Due to the reporting standards of the PLS-SEM method, we are unable to include additional color figures for data visualization. However, we have incorporated color enhancements into the analytical process chart to improve the visual presentation of the article, as per your suggestion.
Comment 4: The emerging studies about immune mediated inflammatory diseases also should introduced, such as doi: 10.1186/s40001-024-01836-1. (Monteleone et al., 2023)
Answer 4: Thank you for your insightful suggestion. We agree that discussing advances in IMID treatment would be a valuable addition. However, instead of the suggested reference, we have cited a more relevant source ([12] doi: 10.1016/j.autrev.2023.103410) that better aligns with our article's focus. The newly added text can be found on page 2 (paragraph 1, line 51-57):
"Despite advances in treatment, managing IMIDs remains challenging due to variations in disease mechanisms and individual responses to therapy. Emerging research highlights that while IMIDs share common inflammatory pathways, differences in cytokine function and immune regulation contribute to variability in treatment effectiveness [12]. Many individuals remain unresponsive or intolerant to current therapies, underscoring the need for novel approaches that target multiple inflammatory pathways simultaneously."
Comment 5: There are some grammatical and spelling errors in the paper. Please carefully review the manuscript and make the necessary corrections.
Answer 5: Thank you for your careful review. We have meticulously revised the manuscript for grammar and spelling accuracy. A highlighted version of the revised manuscript has been submitted as a supplementary file for your review.
Reviewer 2 Report
Comments and Suggestions for Authors
Please see my comments below:
- Please ensure that numerical values are consistently represented throughout the manuscript, for example, use "0.79" instead of ".79."
- There are numerous grammatical and language-related errors throughout the manuscript. I recommend thoroughly reviewing and proofreading the entire paper for grammatical accuracy. For example, in line 191, "IBD sample model" would be the correct phrasing. A comprehensive English edit will enhance readability and improve clarity for the readers.
- Were p-values calculated for the variables presented in Table 1? If so, please include them in the table.
There are numerous grammatical and language-related errors throughout the manuscript. I recommend thoroughly reviewing and proofreading the entire paper for grammatical accuracy. For example, in line 191, "IBD sample model" would be the correct phrasing. A comprehensive English edit will enhance readability and improve clarity for the readers.
Author Response
Comment 1: Please ensure that numerical values are consistently represented throughout the manuscript, for example, use "0.79" instead of ".79."
Response 1: Thank you for identifying this inconsistency. We have carefully reviewed the manuscript to ensure uniform representation of numeric values. The only instance we found using "0." was on page 4, paragraph 2, line 151, which has now been corrected to:
"Significance levels were assessed using p-values, considered significant if below .05."
Comment 2: There are numerous grammatical and language-related errors throughout the manuscript. I recommend thoroughly reviewing and proofreading the entire paper for grammatical accuracy. For example, in line 191, "IBD sample model" would be the correct phrasing. A comprehensive English edit will enhance readability and improve clarity for the readers.
Response 2: We appreciate your attention to detail. As suggested, we have adopted the terminology "IBD sample model" (page 6, paragraph 4, line 202) and ensured consistency by using similar phrasing for other models (e.g., "The PsO sample model" on page 6, paragraph 5, line 211). Additionally, we have revised the text throughout the manuscript to maintain consistency in terminology. A revised version with all modifications highlighted has been submitted as a supplementary file.
Comment 3: Were p-values calculated for the variables presented in Table 1? If so, please include them in the table.
Response 3: Thank you for your suggestion. We would like to clarify that the demographics table serves as a descriptive overview of the sample characteristics rather than a comparative analysis between groups. Since the table is not intended for hypothesis testing or statistical comparisons, p-values are not typically included. Instead, its primary purpose is to present the distribution of key demographic variables—such as age, gender, and education level—to provide the reader with a comprehensive understanding of the study population.
Comment 4: There are numerous grammatical and language-related errors throughout the manuscript. I recommend thoroughly reviewing and proofreading the entire paper for grammatical accuracy. For example, in line 191, "IBD sample model" would be the correct phrasing. A comprehensive English edit will enhance readability and improve clarity for the readers.
Response 4: Thank you for your feedback. A revised version of the manuscript, with all modifications highlighted, has been submitted as a supplementary file for your review.